# Phytochemical Screening, Antioxidant and Antibacterial Properties of Extracts of *Viscum continuum E. Mey. Ex Sprague*, a South African Mistletoe

**DOI:** 10.3390/plants11162094

**Published:** 2022-08-12

**Authors:** Sipho Mapfumari, Noel-David Nogbou, Andrew Musyoki, Stanley Gololo, Mmamosheledi Mothibe, Kokoette Bassey

**Affiliations:** 1Department of Pharmaceutical Sciences, School of Pharmacy, Sefako Makgatho Health Sciences University, P.O. Box 218, Pretoria 0208, South Africa; 2Department of Microbiology, School of Medicine, Sefako Makgatho Health Sciences University, P.O Box 211, Pretoria 0208, South Africa; 3Department of Biochemistry, School of Science and Technology, Sefako Makgatho Health Sciences University, P.O Box 235, Pretoria 0208, South Africa; 4Department of Pharmacology, Faculty of Pharmacy, Rhodes University, P.O. Box 94, Grahams Town 6139, South Africa

**Keywords:** Mistletoe, antioxidant, antimicrobial, bio-autography

## Abstract

*Viscum continuum E. Mey. Ex Sprague* is a woody evergreen semi-parasitic shrub that grows on the branches of other trees. It is used by African traditional healers for post-stroke management. This study reports on the qualitative phytochemical screening and the antioxidant and antimicrobial activities of *Viscum continuum’s* acetone, methanol, hexane and dichloromethane extracts. Standard protocols for the phytochemical screening of extracts were employed. TLC bio-autography was used for qualitative antioxidants analysis. Assays: 2,2-diphenyl-1-picrylhydrazyl, H_2_O_2_ free-radical scavenging and ferric chloride reducing power were carried out for quantitative antioxidant analysis. The antimicrobial potential of extracts was screened using disc diffusion, bio-autography and broth micro-dilution. The results indicate the presence of alkaloids, phenolics and tannins in all extracts. Acetone and methanol revealed significant amount of saponins, phenolics, tannins and terpenoids. The extracts exhibited significant antioxidant potential on TLC with positive compound bands at an Rf range of 0.05–0.89. DPPH, H_2_O_2_ and the reduction of Fe^3+^ to Fe^2+^ assays indicated that methanol extract has a strong antioxidant potential, followed by acetone, DCM and lastly hexane. The extracts of *Viscum continuum* show the potential to be antibacterial agents. It can be concluded that *Viscum continuum* extracts contain phytochemicals which are capable of mitigating against chronic health conditions such as cancer, stroke and stress-related and infectious diseases.

## 1. Introduction

In recent years, medicinal plants have taken center stage as a rich natural source of novel drugs. This is because medicinal plants produce secondary metabolites which are bioactive and are considered safe and biodegradable [1,2]. Laboratory-based in vitro studies have associated these compounds with anti-inflammatory, antidiabetic, anticancer and antioxidant properties as well as numerous other pharmacologically important activities [3,4,5,6,7]. Even though the mechanism of the actions of some of the secondary metabolites may not be fully understood, there is a consensus that their structural diversity is what gives them their wide pharmacological applications [8,9]. Alkaloids, anthraquinones, flavonoids, glycosides, saponins, tannins and terpenes appear to be the commonly studied and reported phytochemicals [2,10,11,12]. A number of studies have linked medicinal plants which possess these phytochemicals to biological activities such as antioxidant, antibacterial, anticancer and antidiabetic properties [13,14,15,16]. The biological activity of paramount importance in medicinal plants is antioxidant activity. This is because oxidant stress caused by an excessive amount of oxidants is associated with a number of human ailments which include but are not limited to cancer, the development of neurological disorders and cardiovascular diseases [17,18,19]. It is worth noting that, physiologically, oxidants appear in two forms: as reactive oxygen species or reactive nitrogen species, which can either be free radicals or molecular species capable of generating free radicals [20,21]. A homeostatic balance between the presence of oxidants and antioxidants needs to be maintained, since oxidants play a significant physiological role in insulin synthesis, the regulation of vascular tone and cell proliferation, as well as other metabolic processes [18,20,22,23].

Physiologically, there are inherent mechanisms which play the regulatory role that maintain the homeostatic status quo of oxidants, but supplementary antioxidants, which are mainly provided by medicinal plants, serve as the most natural intervention if the inherent mechanism is not enough [24,25]. Not only are antioxidants from medicinal plants important for maintaining balance, but their metabolism has been reported to produce metabolites which may be crucial in managing as well as treating other ailments [25]. A number of species of Mistletoe trees from other parts of the world, such as *Phoradendron flavescens, Viscum album L*. and *Loranthus micranthus*, have been studied and found to possess these vital phytochemicals [26,27,28]. The South African Mistletoe tree, identified as *Viscum continuum* E.mey. ex sprague by the South African National Biodiversity Institute (SANBI), is amongst the medically important plants which are used by African traditional healers for post-stroke management. It is a woody evergreen semi-parasitic shrub that grows on the branches or trunks of other trees and can stem up to 2 m long. There is insufficient literature information on the phytochemistry, biological activities as well as the indigenous medical uses of the Mistletoe tree of the South African ecotype. As such, this study reports for the first time on the phytochemical composition of *Viscum continuum E.mey. ex sprague* as well as the antioxidant and antimicrobial properties of its hexane, dichloromethane, acetone and methanol extracts.

## 2. Results

### 2.1. Qualitative Phytochemical Screening 

The mass and percentage yield of the Mistletoe tree extract increased using acetone (13.13 g: 1.32%), dichloromethane (19.04 g: 1.90%), hexane (19.23 g: 1.92%) and methanol (23.67 g: 2.37%). Altogether, 75.07 g of dry extract at a percentage yield of 7.51% was derived from 1.05 kg of the dry plant material. All the extracts obtained from the Mistletoe tree sample were subjected to qualitative phytochemical screening using standard chemical tests. The results on whether the targeted phytochemicals were present or not in the plant extract are presented in Table 1.

The phytochemical chemical tests on the Mistletoe tree extracts indicated the presence of alkaloids in all extracts except that of hexane. Given that alkaloids are well-known for their anticancer or antitumor properties, their presence in the plant places it as a prospective source of alkaloids for use in the complementary or alternative treatment of cancer [29,30]. This does not come as a surprise, since related species have already been explored for possible cancer treatments. Although to a varying degree all extracts showed the presence of phenolic compounds as well as tannins, the phenolic compounds are mainly known to be responsible for their antiseptic, anti-inflammatory, antioxidant as well as anti-cancer properties of Mistletoe plants [31,32]. This suggests that all the South African Mistletoe tree extracts should have some degree of antioxidant activity in addition to the other activities. The non-polar extracts showed the presence of flavonoids as well as cardiac glycosides, whereas the polar ones were negative for both. The absence of flavonoids in the polar extracts presents a worrying revelation, since they are known to be powerful antioxidants which provide humans with immune-boosting benefits [33] and which play a role in brain functions and importantly in blood sugar and pressure regulation [3,34]. Terpenoids were present in hexane, acetone and methanol but were absent in the dichloromethane extract. The structural range of terpenoids has been reported to make them the most important of all phytochemicals, since it gives them their diverse biological activities. Amongst the reported biological activities of Mistletoe trees are anticancer [35], antimicrobial [36], hepatoprotective, antimalarial [37] and antiviral [38] effects as well as the inhibition of cholesterol synthesis [39]. The presence of this group of compounds in the South African Mistletoe tree underlines its importance as a plant drug that is used by rural South Africans. As indicated in Table 1, both the polar extracts showed the presence of saponins, and the test indicated the absence of these phytochemicals in the non-polar extracts. This is significant because, in the indigenous practices of decoction preparation, the solvents used are polar in nature, which are water, alcohol or a combination of both. This is particularly important since these phytochemicals have been reported to be responsible for multiple biological activities such as antioxidant, antibacterial as well as anticancer [40,41,42] activities. Hexane and methanol extracts showed the presence of steroids; however, they only appeared as traces in the acetone extract and were totally absent in the dichloromethane extract.

### 2.2. Qualitative Antioxidant Assay by DPPH and TLC Direct Bio-Autography Dot Blot Assay

The results obtained in this study revealed a cream coloration against the purple background on the TLC plates, indicating positive outcomes for antioxidant potentials [15]. The cream coloration was observed for the crude hexane, acetone and methanol extracts. This suggests that all the crude extracts except for the dichloromethane have a compound(s) which can elicit antioxidant activity. It also appeared from the results that more polar extracts led to more intense antioxidant activity, and this could be distinguished by the width and intensity of the zone of inhibition. This is significant because some of the indigenous Mistletoe preparations, when consumed, can provide the user with antioxidant benefits. Developed TLC analysis was used to evaluate if the antioxidant activity of each of the extracts in the dot blot assay was as a result of the synergy of the different phytoconstituents and if inactivity was a result of antagonistic action. As such, the developed TLC spotted with each of the extracts demonstrated that each of the extracts possessed specific compounds responsible for that extract’s antioxidant activity. It is interesting to note that the dichloromethane extract, which, in the dot plot analysis, showed no activity, also exhibited bands on the developed TLC with antioxidant activity. The specific antioxidant compound bands and their respective Rf values are displayed in Figure 1 and are summarized in Table 2.

As evident in Figure 1, not all the phytochemicals present in the dichloromethane extract of the South African Mistletoe tree exhibited antioxidant biological activity. For instance, the bands with Rf values at 0.09, 0.25, 0.49, 0.55, 0.60, 0.63 and 0.81 are visible in plates 1, 2 and 3. However, when compared with plate 4 for antioxidant evaluation, these bands tested negative. It is logical to suggest that the dichloromethane not showing any antioxidant activity from the dot plot analysis implies that the antioxidant activity of the bands at 0.08, 0.12, 0.13, 0.25, 0.38, 0.42 and 0.89 of this extract (plate 4) is not a function of synergy between the compounds represented by these bands.

The antioxidant bands were at Rf values of 0.05 (dark blue), 0.42 (orange) and 0.89 (pink; at 366 nm), which could suggest the presence of anthocyanins, beta-carotene and lycopene, which are all natural antioxidants present in the plant. Another plausible explanation for the negative antioxidant activity result on the dot blot for dichloromethane extract could be sample-color interference. As already stated, there were seven antioxidant active compound bands in the dichloromethane extract, as indicated in Table 3. It is worth noting that the methanol extracts appeared to be entirely capable of eliciting antioxidant activity. The Rf values of all antioxidant compound bands for the acetone and methanol extracts are also displayed in Figure 2 and Table 3. The acetone extract displayed three antioxidant phytochemical bands at Rf values of 0.01, 0.05 and 0.82, and the methanol extract revealed eleven of such antioxidant bands at Rf values of 0.05, 0.10, 0.15, 0.20, 0.25, 0.30, 0.35, 0.40, 0.45, 0.53 and 0.65, making it the most active extract against reactive nitrogen species as represented by DPPH.

### 2.3. Quantitative Antioxidant Assay by DPPH, Hydrogen Peroxide Free-Radical Scavenging Potential and Ferric Chloride (Fe^3+^–Fe^2+^) of South African Mistletoe Extracts

The quantitative DPPH free-radical scavenging activity of the four Mistletoe extracts was evaluated at 517 nm. The decrease in absorbance with an increase in the concentration of the extracts was an indication of the extracts donating protons to the DPPH free radical. However, with respect to the percentage inhibitory activity of the extracts, as shown in Figure 3, the methanol extract exhibited better inhibitory activity of 98% compared to all the analytes. This was followed by the gallic (85%) acid standard, acetone extract (79%), hexane (63%), dichloromethane (45%), butylated hydroxytoluene (BHT, 18%) and the diosgenin standard (Figure 3).

These results are in agreement with those obtained qualitatively, i.e., the antioxidant activity of South African Mistletoe tends to increase as the polarity of the solvent used for extraction increases, such that the more polar methanol extracts in this study had the highest DPPH free-radical activity compared to the acetone, dichloromethane and the hexane extracts. This finding is intriguing considering the fact that the methanol extract was more active than the standard antioxidant compounds used. Furthermore, the H_2_O_2_ free-radical scavenging potential of the South African Mistletoe extracts was very similar to the results obtained for the DPPH scavenging assay. The similarity is based on the observation that the methanol extract also exhibited the best scavenging ability of 98% (Figure 4) from the hydrogen peroxide species.

However, from Figure 4, one can see that the high methanol H_2_O_2_ scavenging activity was followed by the dichloromethane extract at a concentration range of 0.2–0.8 mg/mL, but it normalized to align with the acetone extract, as was the trend for the DPPH assay at 0.8–1.0 mg/mL. A rational explanation to this observation could be that the scavenging of free radicals from the H_2_O_2_ species by the plant extracts is not a function of the solvent polarity. In addition, measurements of the reductive ability of Fe^3^^+^–Fe^2^^+^ transformation in the presence of Mistletoe methanol, acetone, hexane and dichloromethane extracts were also assessed. The reducing capacity of a compound or extract usually serves as a significant indicator of its potential antioxidant activity [43].

In DPPH and H_2_O_2_ free-radical scavenging assays, a high percentage value is an indicator of the extract with the best free-radical scavenging activity. In contrast, the best antioxidant activity for the reduction of Fe^3+^ to Fe^2+^ is normally earmarked by a relatively low percentage value. Therefore, the reducing power assay of the extracts supports the DPPH and H_2_O_2_ assay because the best activity for this assay is from methanol extract at 50%, followed by acetone at 62%. These results still underscore the trend that the antioxidant activity of the South African Mistletoe tree increases as the extract polarity increases. However, it must be mentioned that, although the antioxidant activity of the methanol extract from the DPPH and the H_2_O_2_ assay was better than those of the standard compounds used, the converse is the case except for gallic acid with the reducing power assay (Figure 5). Possibly, the redox reaction mechanism is responsible for the change in the trend of antioxidant activity. Overall, the IC_50_ data in Table 4 summarizes the trend in better antioxidant activity from methanol (IC_50_ = 0.11 mg/mL), acetone (IC_50_ = 0.33 mg/mL), hexane (IC_50_ = 0.64 mg/mL) and dichloromethane (IC_50_ = 0.95 mg/mL) for the DPPH assay. A similar proportionality in the trend of antioxidant activity was observed for the H_2_O_2_ and reduction of Fe^3+^–Fe^2+^ analysis of the extracts, as mentioned earlier.

### 2.4. Qualitative Antimicrobial Assay by Direct Bio-Autography Dot Blot Assay of Mistletoe Tree Extracts

For consistency, the bio-autography assay was employed to view the antimicrobial potentials of the four Mistletoe extracts immediately. The results obtained (Figure 6) display control plate 1 compared with those under antimicrobial investigations (plates 2–5). Of the four Mistletoe extracts under antimicrobial investigation, only the acetone and the methanol extracts revealed potentials for antimicrobial activities against *S. sense* clinical isolate (Figure 6 plate 2), judging by the cream coloration against the brownish background [44]. This preliminary antimicrobial activity of the methanol and acetone extracts against *S. sinensis* is in line with the qualitative and quantitative antioxidant activity of the extracts. The apparent inactivity of the hexane and dichloromethane extracts against all the test organisms investigated in this study and that of the acetone and methanol against *B. subtilis*, *E. coli* and *K. pneumonia*, as previously mentioned, may be linked to the zero synergy between the phytoconstituents in the plant extract. Possibly, there may be some sort of activity against the test organism if the phytochemicals are to act against these organisms independently. Part of the focus of this study was to investigate for any scientific proof to authenticate the traditional uses of Mistletoe in the management of bacterial infections by the indigenous people of South Africa. As a result, clinical isolates were used for the TLC direct bio-autography assay to assess the antimicrobial potentials of the plant extracts. Each extract was spotted on the TLC plate pre-coated with silica and developed using a suitable solvent. The plates were then sprayed with overnight cultures of *E. faecalis, S. sinensis, B. subtilis* and *A. baumannii* broth solutions.

In Figure 6, there is a compound band at an Rf value of 0.23 in the acetone extract that is active against *A. baumani* (plate 2) more than in the other extracts. In plate 3, however, there are over 2-3 compound bands in the hexane and dichloromethane extracts that indicate inhibition against *S. sinensis*, and in plate 4, there is a compound band at an Rf value of 0.49 in the hexane and acetone extracts with possible activity against *E. faecalis.* The hexane and dichloromethane extracts display a common compound band at Rf = 0.25 that is active against *S. substilis*. This compound appears to be present in the acetone extract as well but exhibits less activity judging by the discoloration. It is enthralling to note that the methanol extract of the South African Mistletoe tree does not show any sign of activity even to the strain of *B. subtilis* that is the most susceptible to the other three extracts as well as the *A. baumannii* clinical isolate from the bio-autography assay. This can be attributed to the activity of this extract being associated with synergism action because the same extract was active from the disc diffusion assay. Regarding the disc diffusion results, the acetone and methanol extracts showed activity against *A. baumannii* BAA747. This can be attributed to the activity of this extract being associated with synergism action because the same extract was active from the disc diffusion study. Regarding the disc diffusion results, the acetone and methanol extracts showed activity against *A. baumani* BAA747. This can be seen by the clear zone of inhibition around the discs of both the acetone solvent and the methanol solvent, contrary to what was observed for the TLC dot blot assay. Comparing all the extracts with the Acetone-DMSO and Methanol-DMSO controls, one can suggest that all the extracts are inactive against the *A. baumannii* clinical isolate and *K. pneumonia* ATCC 25922. The *A. baumannii* BAA747 strain appears to be the most susceptible to the polar acetone and methanol extracts, and *E. coli* is partially susceptible to these extracts.

### 2.5. Quantitative Antimicrobial Evaluation by MIC Assay

A study in 2017 by Upadhayay and colleagues suggested that, in order to align the laboratory-based results and to consider the results clinically significant, the minimum inhibitory concentration for a plant extract can be capped at 1.00 mg/mL [45]. As such, the antibacterial work that was performed in this study was carried out with a maximum concentration of 1mg/mL, which was the same concentration as the standards used. The antibacterial properties of the plant extracts are summarized in the table below.

The results of this study suggest that *Viscum continuum E.mey. ex sprague* has, to a certain extent, some antibacterial activity as can be observed from Table 5. This can mainly be seen in the DCM and acetone extracts, where, under disc diffusion and bio-autography, compounds with specific Rf values are responsible for the antibacterial activities. The MIC values suggest that DCM (0.75) is the only extract that falls below the maximum (1mg/mL) used for the 6245 strain of A. baumani. With an equal value of 0.75 mg/mL, the DCM and acetone extracts were the only extracts which showed significant antibacterial activity against *E. coli*. All the extracts showed inhibition below the 1mg/mL maximum limit set for this study, with their MIC ranging between 0.375 and 0.75, as can be observed from Table 5. These results are significant because the MIC was below 1mg/mL even though the plant material used was crude, which implies that it constituted compounds which may have had some antagonistic effect against the active compounds. Since only certain compounds are responsible for the observed activity, further studies are needed to isolate and characterize these compounds.

## 3. Discussion

The importance of understanding the phytochemical profile of a medicinal plant can never be overstated. This study, through the use of standard phytochemical qualitative methods, reports for the first time the phytochemical profile of *Viscum continuum E.mey. ex sprague*. The results indicate the presence of alkaloids, glycoside, saponins, phenolic compounds, steroids, tannins and terpenes within the different extracts of the Mistletoe under investigation. These compounds can also be found in other species of plants from the same family of Viscum, as per previous reports [46,47,48]. The identified phytochemicals are associated with various biological properties, such as but not limited to antiseptic, antioxidant, antibacterial and anticancer effects, as well as other immune-boosting benefits [31,32,33]. These are important biological properties to be associated with a plant found in a developing country, where about 60% of the population relies on medicinal plants to treat various diseases. As such, the phytochemical profile results of this study places *Viscum continuum E.mey. ex sprague* as a potential source of compounds which can be explored for human health benefits.

This is further supported in this study by the different extracts’ strong antioxidant as well as antibacterial activities. These results are coherent with earlier studies, which have suggested that *Viscum album, Viscum abeitis* and *Viscum autriacum* extracts are strong antioxidant agents [49,50]. The methanolic extracts of these related species are comparable with the outcomes of this study, as they all are in concurrence across the different extracts, and they are extracts with strong antioxidant activities [49]. As previously reported, ingested or supplementary antioxidants generate other bioactive metabolites essential for the treatment of several other ailments [25], besides their main role in oxidative stress reduction. Similar to previous studies, where it was determined that other species from the family Viscum have antibacterial properties against E faecalis, B subtilis, E coli and K pneumonia [51,52], the results of this study indicate that V continuum has comparable properties. This in turn suggests that the South African Mistletoe tree has great potential to treat multiple ailments. The results of this study also position *Viscum continuum E.mey. ex sprague*, the South African Mistletoe tree, as an antibacterial agent. In an era where there is a rise in antimicrobial resistance and in a country where there are poor hygiene practices, this plant has the potential to be the source of antibacterial agents which can be used to reduce death resulting from microbial infections. Based on the phytochemical profile and the confirmed biological potentials of the extracts, this study proposes that South African Mistletoe extracts can mitigate certain microbial infections as well as mitigate oxidant species, thus giving it potential for use in the management of infectious and oxidative-stress-related diseases.

## 4. Materials and Methods

### 4.1. Sample Collection, Preparation and Storage (Plant Material)

The whole shrub of the South African Mistletoe tree was collected on 22 July 2019 at Mmabatho (25.8378° S, 25.5948° E), which is situated in Molopo, North West province, South Africa. This was conducted under the guidance of an indigenous knowledge systems (IKS) practitioner. The shrub was then taken to the South African National Biodiversity Institute (SANBI), where it was identified as *Viscum continuum E.mey. ex Sprague*, and a voucher number (MTT01) was allocated prior to depositing the sample in the herbarium of the School of Pharmacy of Sefako Makgatho Health Sciences University (SMU). The plant sample was dried at room temperature and was then ground to powder using Polymix Laboratory Dry Mill Drive Unit (Polymix™ PX-MFC 90 D, Kinematica AG, Luzern, Switzerland) at 3500 revolutions per minute (RPM), and the particle size was kept consistent using the sieve that comes standard with the instrument. The resultant powder was stored in the dark until needed for further processing.

### 4.2. Extraction of the Plant Material

All the solvents used in the study were purchased from Rochelle Chemicals and Lab Equipment Cc, Johannesburg, South Africa. The ground sample (1.05 kg) of the plant was extracted using NUVE shaking water bath (ZT10.ST 30, KK05F01, Akyurt, Turkey) at 100 RPM with 2.5 L of n-hexane, dichloromethane, acetone and methanol in a serial exhaustive cold maceration procedure. This was performed in a 24 h cycle for each run. The extracts which resulted from the process were then filtered and concentrated using a Stuart rotary evaporator (Re400, Cole-Parmer Ltd. Stone, Staffordshire United Kingdom) and were allowed to dry fully under a stream of air. The total mass of each of the extracts obtained was weighed, and the percentage yield was calculated.

### 4.3. Qualitative Phytochemical Screening

In order to test for the presence of different phytochemicals in the extracts of the South African Mistletoe tree, standard chemical tests for alkaloids [53], flavonoids [54] glycosides [55,56], phenolics [57], terpenoids [58], tannins and saponins [2] as well as steroids [59] were employed with minor modifications.

### 4.4. Selected Biological Activity Assay

#### 4.4.1. Qualitative and Quantitative Antioxidant Assay

The four extracts (hexane, dichloromethane acetone and methanol extracts) were reconstituted by dissolving 1.00 mg of each extract in 1.00 mL of hexane, dichloromethane acetone and methanol. A thin-layer chromatography analysis was conducted by spotting 5.00 µL of the extract solution at the base, which was drawn at 1.5 cm from the bottom of the aluminum pre-coated F_254_ plates and which was allowed to develop with the use of a different solvent system for the different extracts. The hexane extract was developed with hexane: ethyl acetate (9:1 *v/v*), the dichloromethane extract was developed with hexane: ethyl acetate (7.5: 2.5 *v/v*) and the acetone and methanol extracts were developed with ethyl acetate: methanol: water (8:4:1 *v/v/v*). The developed plates were each then observed under normal light for visible bands and then visualized under a UV lamp under 254 nm and 366 nm. The plates were thereafter sprayed with 0.4 mM solution 2,2-diphenyl-1-picrylhydrazyl (DPPH) dissolved in methanol. The formation of a yellow to cream spot(s) against the purple background was taken as a positive indication for the presence of antioxidant compounds.

#### 4.4.2. DPPH Free-Radical Scavenging Activity of the Mistletoe Extracts

This was achieved following a method described by [60,61], with some slight modifications. To that effect, a range of concentrations (0.2 mg/mL to 1.0 mg/mL) of the non-polar hexane and dichloromethane and the polar acetone and methanol extracts were prepared. A DPPH solution of 0.2 mg/mL was also prepared, and 1.0 mL of this solution was mixed with 1.0 mL of the extract solution in a test-tube. The contents in the test-tube were vortexed to thoroughly mix and were then placed in dark cardboard for 30 min. The absorbance of the different concentrations was spectrophotometrically measured at 517 nm using a 96-well microplate-reader spectrophotometer (SprectraMax^®^, Molecular Devices, California, United States). The same concentration of gallic acid (a phenolic compound standard), diosgenin (a saponin compound standard) and butylated hydroxyl toluene (BHT) (a synthetic antioxidant compound standard) were used as reference standards. The percentage radical scavenging activity of the extracts was calculated using the equation below:(1)$$\%\mathrm{DPPH}\;\mathrm{radical}\;\mathrm{scavenging}\;\mathrm{activity}=\frac{A_{0}-A_{s}}{A_{0}}\times 100$$
where A_0_ is the absorbance of the negative control and A_s_ is the absorbance of the extracts/standards.

#### 4.4.3. Ferric Chloride Reducing Power Assay

The method used was adopted from [60] and was carried out with slight modifications. The different extracts were re-dissolved into the solvents, which they were originally extracted with. Then, different concentrations from 0.2 mg/mL to 1.0 mg/mL were prepared. Each of these was mixed in a test-tube with 2.5 mL of 0.2 M phosphate buffer (pH 6.6) and 2.5 mL of 1% (*w*/*v*) of potassium ferricyanide (K_3_Fe(CN)_6_). After mixing the contents in the test-tube, they were incubated for 20 min at 50 °C followed by the addition of 2.5 mL trichloroacetic acid (10% *w*/*v*), and they were centrifuged for 10 min at 3000 rpm. A volume of 2.5 mL of the upper layer of the solution was mixed with 2.5 mL of distilled water and 0.5 mL of ferric chloride (0.1% *w*/*v*). The absorbance of the mixture was measured at 700 nm using the spectrophotometer. The procedure was repeated for gallic acid, diosgenin and butylated hydroxyl toluene, which were used as reference standards. The percentage reducing power of the extracts was determined using the equation below:(2)$$\%\;\mathrm{Reducing}\;\mathrm{power}=\frac{A_{0}-A_{s}}{A_{0}}\times 100$$
where A_0_ is the absorbance of the negative control and A_s_ is the absorbance of the extracts/standards.

#### 4.4.4. Hydrogen Peroxide Scavenging Assay

In order to assess the hydrogen peroxide scavenging potential of the Mistletoe tree extracts, the method described by [62] was followed with minor modifications. A volume of 2 mL hydrogen peroxide (20 mM), which was prepared in a phosphate buffer saline (pH 7.40), was mixed with 1.0 mL of each of the different concentrations (0.2 mg/mL to 1.0 mg/mL) of the extracts from the stock solutions. These were then mixed thoroughly with the use of a vortex and were then incubated for 10 min before their absorbance was measured at 560 nm using the spectrophotometer. The reference standards for this assay were 1.00 mg/mL each of gallic acid, diosgenin and butylated hydroxyl toluene. The percentage hydrogen peroxide inhibition ability of the extracts was calculated using the equation below:(3)$${\%\;H}_{2}O_{2}\;\mathrm{inhibition}\;=\frac{A_{0}-A_{s}}{A_{0}}\times 100$$
where A_0_ is the absorbance of the negative control and A_s_ is the absorbance of the extracts/standards.

### 4.5. Qualitative and Quantitative Antimicrobial Assay

#### 4.5.1. Test Organisms

The microorganisms used in this study were Acinetobacter *baumannii (A. baumannii*) BAA747; *Pseudomonas aeruginosa (P. aeruginosa)* ATCC 27853; *Klebsiella pneumoniae (K. pneumoniae)* ATCC 25922; *Enterococcus faecalis* (ATCC 51299); and four clinical isolates, namely: *A. baumannii 6245; Bacillus subtilis ATCC19659 (B. subtilis); Streptococcus sinensis (S. sinensis);* and *Escherichia coli ATCC35218 (E. coli).*

#### 4.5.2. Antibacterial Susceptibility Testing

To determine the antibacterial potential of the different extracts of *Viscum continuum E. mey. Ex Sprague*, the Kirby–Bauer disc diffusion method [63] was adopted with some modifications. A concentration of 1.0 mg/mL [45] of each of the four extracts was prepared using 10% DMSO, and another 1.0 mg/mL was prepared using the respective solvents, which were used for extraction in a sterile 2.0 mL screw cap tube. Discs were punched from sterile filter paper (Whatman No. 1), sterilized and soaked on each of the tubes containing the different extracts. After being standardized to 0.5 Mac Farhland using a turbidity meter (DensiCHEK, BioMerieux, Marcy-I’Etoile, France ), the test organisms were lawned on to the Mueller–Hinton (MH) agar plates, and each organism had its own agar plate. A paper disc laced with each of the different extracts was placed on the same plate as the controls. Depending on the test organism being assessed, different controls were used: *A. baumanni* (imipenem), *P. aeruginosa* (ceftazidine), *B. subtilis* (piperacillin/tazobactam), *S. sinensis* (ceftriaxone) and *K. pneumoniae* (ampicillin). The plates were then incubated for 24 h at 37 °C. The formation of clear zones around the discs was considered as a positive indication that a particular extract was potent against the test organism, and wider zones indicated more effectiveness of the plant extract.

#### 4.5.3. Bio-Autography

To assess for specific compounds which are responsible for the identified antibacterial activity, the standard bio-autography method, as outlined by [64], was performed with modifications. TLC plates of the different extracts were analyzed by spotting each of the extracts at 1.5 cm from the base of the plate and were allowed to develop in different solvent systems, and they were taken out when the solvent front had reached at least 8 cm from the base of the paper. The different test organisms were cultured overnight in MH broth, and their growth was adjusted to 0.5 Mac Farhland (1.5 × 10^8^ colony forming per milliliter (CFU/mL)). The developed TLC plates were sprayed with the bacterial suspension until fully wet. This whole process was carried out in a laminar flow cabinet (Labotec, SA). Thereafter, the plates were incubated overnight at 37 °C and 100% relative humidity in the dark. After incubation, the plates were sprayed for visualization with 2.0 mg/mL solution of *p*-iodonitrotetrazolium (INT) violet (Sigma-Aldrich) indicator and were further incubated overnight. A white to yellowish color against the purple background was considered a positive indication that a particular compound(s) band exerted an inhibitory effect against the test organism. The Rf values for all the compound bands which showed activity against the different test organisms were recorded.

#### 4.5.4. Minimum Inhibition Concentration (MIC)

In determining the lowest concentration of the extract required to have an antagonistic effect on the growth of test organisms, the broth micro-dilution method described by [44] was used. All different extracts were prepared at a final concentration of 1.0 mg/mL. A volume of 150 µL of MH broth was added to all the wells on the 96-well plates. For the first well of each row, 150 µL of the plant extract was added, followed by serial two-fold dilution. The test organism (50 µL) at 1.5 × 10^8^ (CFU/mL) was added to each well, followed by an overnight incubation at 37 °C. After that incubation, 2 drops of 0.2 mg/mL INT were added to all the wells in order to reveal any bacterial growth. The lowest concentration that inhibited the growth of the bacteria was taken as the MIC for the particular extract against the test organisms.

## 5. Conclusions

This study, through the use of standard phytochemical qualitative methods, has shown that *Viscum continuum E. mey. ex Sprague*, which is a South African Mistletoe ecotype, contains alkaloids, glycoside, saponins, phenolic compounds, steroids, tannins and terpenes. Given the reported biological activities associated with these compounds, the results of this study places *Viscum continuum E. mey. ex sprague* as a potential source of compounds which can be used for human health benefits. This is further supported by the extracts’ strong antioxidant as well as antibacterial activities, displayed by the constituents of the DCM, acetone and methanol extracts. Based on these biological potentials of the extracts, this study proposes that the South African Mistletoe extracts can mitigate certain microbial infections as well as reductions in reactive oxidant species, thus giving it the potential for use in the management of oxidative-stress-related diseases as well as infectious diseases.

## Figures and Tables

**Figure 1 plants-11-02094-f001:**
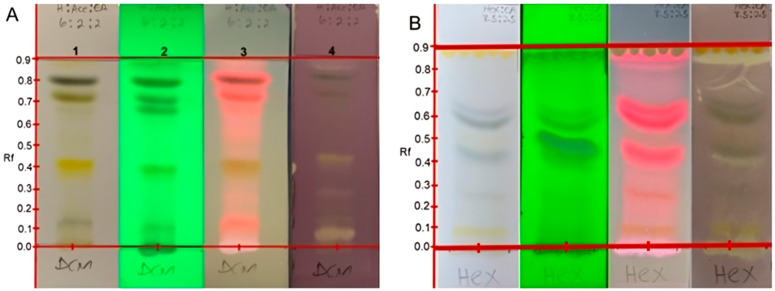
TLC chromatogram of the dichloromethane (**A**) and hexane (**B**) extracts of Mistletoe indicating different phytochemical (plate 1–3) and antioxidant (plate 4) bands. Plates visualized and snapped at white light (1), 255 nm (2), 366 nm (3) and sprayed with 0.2 mM DPPH solution.

**Figure 2 plants-11-02094-f002:**
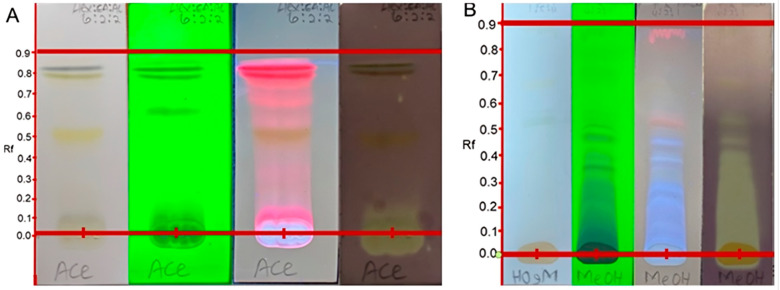
TLC chromatogram of the acetone (**A**) and methanol (**B**) extracts of Mistletoe indicating different phytochemical bands. Plates visualized and snapped at white light (1), 255 nm (2), 366 nm (3) and sprayed with 0.2 mM DPPH solution (4).

**Figure 3 plants-11-02094-f003:**
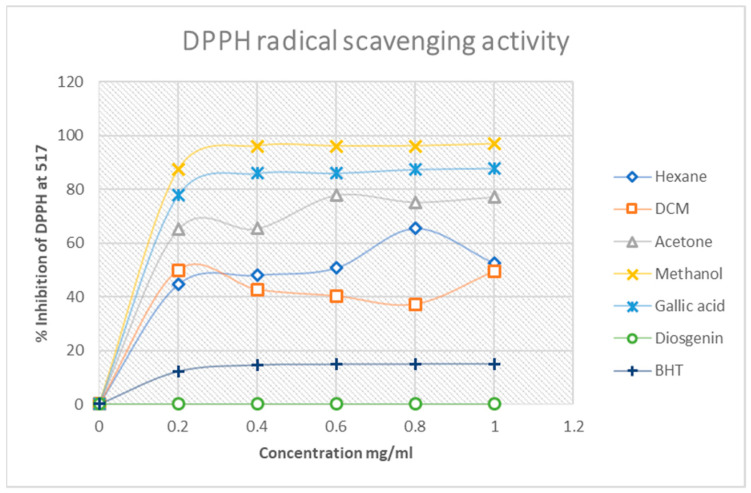
DPPH free-radical scavenging percentage inhibition of Mistletoe extracts and standards. Each value is expressed as mean ± standard deviation (*n* = 3).

**Figure 4 plants-11-02094-f004:**
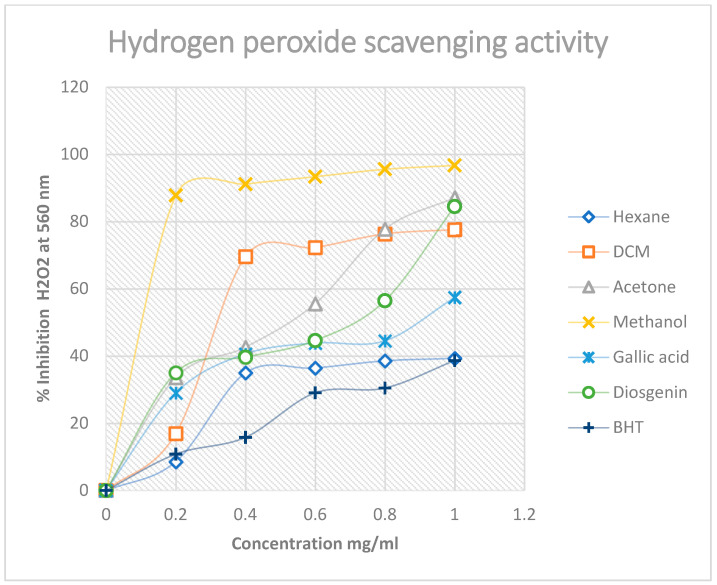
Hydrogen peroxide percentage inhibition of Mistletoe extracts and standards. Each value is expressed as mean ± standard deviation (*n* = 3).

**Figure 5 plants-11-02094-f005:**
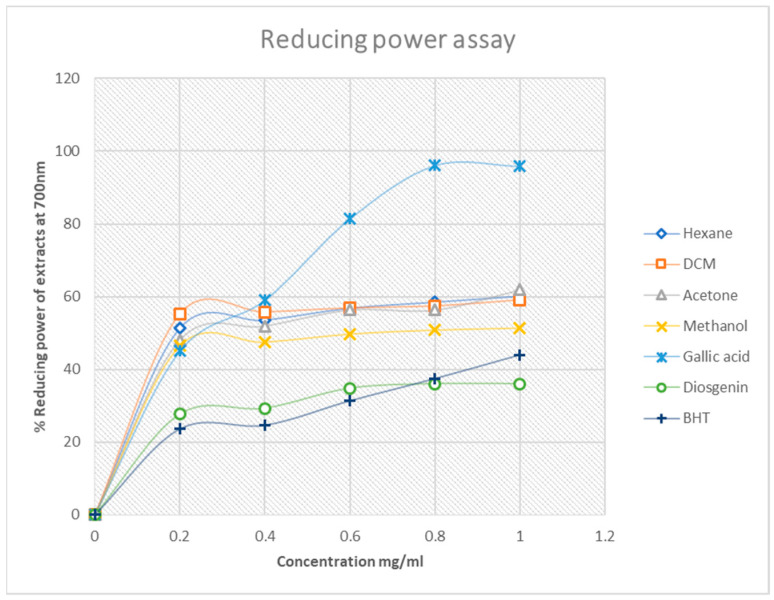
Percentage reducing power of Mistletoe extracts and standards. Each value is expressed as mean ± standard deviation (*n* = 3).

**Figure 6 plants-11-02094-f006:**
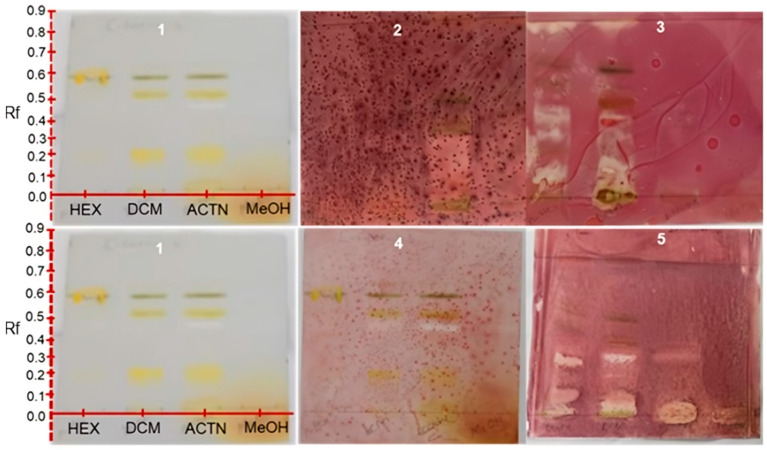
TLC direct bio-autography for the detection of antibacterial activity in 1 = Mistletoe tree extracts (control) antimicrobial activity against *A. baumani* BAA747 (plate 2), *S. sense* (plate 3), *E. faecalis* (plate 4) and *B. substillis* (plate 5). White coloration against purple background indicates inhibition against the test organism.

**Table 1 plants-11-02094-t001:** Qualitative phytochemical present or absent in extracts of the South African Mistletoe tree.

Phytochemicals	Mistletoe Extracts			
	Hexane	Dichloromethane	Acetone	Methanol
Alkaloids	+/−	+	+	++
Glycosides	+	+	−	−
Saponins	−	−	++	+++
Phenolic compounds	+	+	+++	++
Steroids	+	−	+/−	+
Tannins	++	+	+++	++++
Terpenoids	+	−	++	++

**Table 2 plants-11-02094-t002:** Phytochemical bands and their antioxidant potentials in the non-polar hexane and dichloromethane extracts of Mistletoe tree.

	Mistletoe Tree Extracts			
Band No.	Hexane		Dichloromethane	
	Rf-Value	Antioxidant	Rf-Value	Antioxidant
1	0.05	++	0.08	+++
2	0.09	_	0.12	++
3	0.25	_	0.13	+
4	0.42	++	0.15	−
5	0.49	−	0.25	++
6	0.55	−	0.38	+
7	0.60	−	0.42	++
8	0.63	−	0.68	−
9	0.89	+++	0.70	−
10	Nc	Nd	0.81	−
11	Nc	Nd	0.88	+

+++: Significantly present, ++: moderately present, +: sparingly present, −: absent, Nd: not detected and Nc: not calculated.

**Table 3 plants-11-02094-t003:** Phytochemical bands present in the polar acetone and methanol extracts of Mistletoe tree.

	Mistletoe Tree Extracts			
Band No.	Acetone		Methanol	
	Rf-Value	Antioxidant	Rf-Value	Antioxidant
1	0.01	++	0.05	+++
2	0.05	++	0.10	+++
3	0.50	−	0.15	+++
4	0.60	−	0.20	+++
5	0.68	−	0.25	+++
6	0.79	−	0.30	+++
7	0.81	−	0.35	+++
8	0.82	+	0.40	+++
9	Nc	Nd	0.45	+++
10	Nc	Nd	0.50	−
11	Nc	Nd	0.53	+++
12	Nc	Nd	0.65	+
13	Nc	Nd	0.87	−

+++: Significantly present, ++: moderately present, +: sparingly present, −: absent, Nd: not detected and Nc: not calculated.

**Table 4 plants-11-02094-t004:** IC_50_ values indicating the potential inhibition by the four Mistletoe extracts.

Extracts and Standards	IC_50_ (mg/mL)		
	DPPH Scavenging	H_2_O_2_ Scavenging	Ferric Chloride Reducing Power
Hexane	0.64	>1	0.57
DCM	0.95	0.47	0.56
Acetone	0.33	0.51	0.59
Methanol	0.11	0.12	0.73
Gallic acid	0.19	0.79	0.36
Diosgenin	Nd	0.59	>1
BHT	>1	>1	>1

Nd: not determined due to a meager amount of diosgenin standard.

**Table 5 plants-11-02094-t005:** Summary of the antibacterial properties of the extracts of *Viscum continuum E.mey. ex Sprague.*

Bacteria Species	Antibacterial Analysis	Extracts of Solvents with Differing Polarities			
		n-Hexane	Dichloromethane	Acetone	Methanol
6245	Disc diffusion	X	X	X	X
	Bio-autography R_f_ value	-	-	0.77	-
	MIC (mg/mL)	>1	0.75	>1	>1
*E. coli*	Disc diffusion	X	√	√	X
	Bio-autography R_f_ value	-	0.68	0.68	-
	MIC (mg/mL)	>1	0.75	0.75	>1
*A baumani*	Disc diffusion	X	√	√	√
	Bio-autography R_f_ value	-	0.74	0.74	-
	MIC (mg/mL)	0.375	0.375	0.75	0.75
*K pneumonia*	Disc diffusion	X	X	X	X
	Bio-autography R_f_ value	0.00	0.00	0.00	0.00
	MIC	0.75	0.375	0.375	0.75

## Data Availability

The data used in the current study are contained within the article.

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
