# Peer review of "Phytochemical Screening, Antioxidant and Antibacterial Properties of Extracts of Viscum continuum E. Mey. Ex Sprague, a South African Mistletoe"

_plants, 2022, doi:10.3390/plants11162094_

Round 1

Reviewer 1 Report

Some stylistic corrections needed, e.g.: 37 “are are”; 76 “this thus”; 86 “their reported biological -activities of mistletoe tree are anticancer”; 93 “e be it water”; 102 “This then”; 133 beta carotene, probably; 262 “All antibacterial results have summarized on the table below”; Mistletoe is written with small and capital letters – unification needed; ml must be replaced with mL in the whole text; All bacterial latin names must be italic;

143 – what kind of “oxidative species”?

87 – hepaticidal or hepatoprotective?

Figure 3 - A full stop is missing in the caption.

Figure 4 – the caption - Hydripogen? A full stop is missing in the caption.

Figure 5 – what could be explanation of the rapid increase of reducing power at concentrations higher than 0.4 mg/ml for the methanolic extract. It is visible it behaves differently than the other extracts.

218 – “all the test organism – which organisms do authors mean?

Discussion requires improvement – citations about other related plants with similar activities – antioxidant and antimicrobial (comparative discussion with relevant citations are needed); 284 – “This plant which is a South African Mistletoe contains alkaloids, glycoside, saponins, phenolic compounds, steroids, tannins and terpenes” – a citation is needed; 289 – “black population” – do only indigenous black population use folk medicine. Probably other nationalities, that live on these territories also use local folk medicinal remedies.; 293 “The antioxidant results of this study furthermore position this plant as a strong viable option to be explored 294 for human ailment treatment. “ – this idea is repeated in the text.;

Methods – please, explain what kind of controls are diosgenin, butylated hydroxyl toluene.

Do bacterial clinical isolates have a number and/or origin and/or a number under which they are stored.

405 - (Drew, Barry, O'Toole, R. & Sherris, 1972) and (Upadhayay, Singh, & Bhupendra, 2017) – are these citations? If yes – they should me unified with the citation style required. If no – what these brackets represent?

4.5.5 Bio-autography – what is the concentration of cells in the suspension?

4.5.4 MIC- what is the number of cells/mL in the suspension

Conclusions – which are the substances with antioxidant and antimicrobial activity should be mentioned.

Abstract should be adjusted according to the corrections in the main text.

Author Response

Dear Reviewer

Please receive the attached, point-by-point response to your comments on the manuscript we have submitted. The actual changes were made to the manuscript.

Sincerely 

Reviewer 2 Report

The work entitled “Phytochemical screening, Antioxidant and Antibacterial properties of Extracts of Viscum continuum E.Mey. Ex Sprague, a South African Mistletoe” reports on qualitative phytochemicals screening, antioxidant, and antimicrobial activities of Viscum continuum’s extracts obtained using different organic solvents. Extracts revealed significant antioxidant effects and antibacterial action. The subject should be better introduced. Introduction is a bit superficial and should be improved with more detailing. The novelty should be emphasized. The collected data is well presented, even though there are too many figures/tables. However, the discussion does not say much. It would benefit from criticism and more support from the literature. Regardless, the work is noteworthy, of interest and the subject is very pertinent. Data is scientifically sound and was given to the readers in a very conscious manner, for a clear comprehension. The methodology is also sufficiently detailed for a prospective replication to be conducted without complications. Overall, the work has merit and should be considered for publication after these small things are fixed.

Author Response

(The authors gave the same response as above.)

Round 2

Reviewer 2 Report

The authors have addressed all of the reviewers' comments/suggestions and the manuscript is now ready for publication.